# Time-Resolved Proteomics of Germinating Spores of *Bacillus cereus*

**DOI:** 10.3390/ijms232113614

**Published:** 2022-11-06

**Authors:** Xiaowei Gao, Bhagyashree N. Swarge, Winfried Roseboom, Peter Setlow, Stanley Brul, Gertjan Kramer

**Affiliations:** 1Molecular Biology & Microbial Food Safety, Swammerdam Institute for Life Science, University of Amsterdam, 1098XH Amsterdam, The Netherlands; 2Laboratory for Mass Spectrometry of Biomolecules, Swammerdam Institute for Life Science, University of Amsterdam, 1098XH Amsterdam, The Netherlands; 3Molecular Biology and Biophysics Department, UConn Health, Farmington, CT 06030, USA

**Keywords:** *Bacillus cereus*, germination, proteomics

## Abstract

*Bacillus cereus* is a spore-forming human pathogen that is a burden to the food chain. Dormant spores are highly resistant to harsh environmental conditions, but lose resistance after germination. In this study, we investigate the *B. cereus* spore proteome upon spore germination and outgrowth so as to obtain new insights into the molecular mechanisms involved. We used mass spectrometry combined with co-expression network analysis and obtained a unique global proteome view of the germination and outgrowth processes of *B. cereus* spores by monitoring 2211 protein changeovers. We are the first to examine germination and outgrowth models of *B. cereus* spores experimentally by studying the dynamics of germinant receptors, other proteins involved in spore germination and resistance, and coat and exosporium proteins. Furthermore, through the co-expression analysis of 1175 proteins identified with high quality data, germination proteome data were clustered into eight modules (termed black, blue, brown, green, red, turquoise, grey, and yellow), whose associated functions and expression profiles were investigated. Germination related proteins were clustered into blue and brown modules, the abundances of which decreased after finishing germination. In the brown and blue we identified 124 proteins that could be vital during germination. These proteins will be very interesting to study in future genetic studies regarding their function in spore revival in *B. cereus*.

## 1. Introduction

Spore germination followed by outgrowth is an orderly series of physiological and morphogenetic processes initiated by the activation of germinant receptors (GRs) by low molecular weight nutrients. Through this process, spores move from a dormant, resistant state to a vegetative cell that, in the case of *Bacillus cereus,* may produce toxins. The stress resistant spores are generally formed when cells of Bacilli and Clostridia lack nutrients. The resistance of dormant spores to high temperature, UV radiation, desiccation, and various other environmental stresses is a result of their molecular structure. The structure of *B. cereus* spores, from inside to outside, consists of the spore core, inner membrane, the germ cell wall, the cortex, the outer membrane, the coat, and the exosporium. Different from the most well studied spore forming organism, *B. subtilis*, the outmost layer of *B. cereus* spores is the exosporium, a layer that is absent in *B. subtilis*. The relatively dehydrated core characterized by high levels of Ca^2+^-dipicolinic acid (CaDPA) and the saturation of DNA with small acid soluble proteins (SASPs) is crucial for environmental stress resistance. The low permeability inner membrane (IM) and proteinaceous coat layers also contribute to the resistance properties of spores [1].

The tentative model of the spore germination of Bacilli has two main stages (Figure 1a). In stage I, germination is triggered by stimulated GRs. This results in a ~ 30% increase in the spore core water content primarily due to CaDPA release and its replacement by water, and beginning the transition of the spore from the bright phase to dark phase. Then, the spores enter into germination stage II, as two cortex-lytic enzymes CwlJ and SleB are activated to depolymerize the cortex, which allows for core expansion and for an increase in its water content to the level of vegetative cells. The cortex lytic enzyme CwlJ is activated by the released CaDPA, while SleB is activated in an, as yet unknown, manner. Cortex hydrolysis in Bacillus requires YpeB-SleB and CwlJ-GerQ, where YpeB is required for SleB localization and function, and GerQ for CwlJ [2]. Full hydration of the spore core is required for metabolism and macromolecular biosynthesis in spore outgrowth. After completion of the outgrowth, spores start actively growing vegetative cells [1,2]. Previously, the dynamics of the protein and mRNA expression during germination were studied in *B. subtilis* [3]. Although the overall model for *B. subtilis* spore germination is thought to describe the process in *B. cereus*, this has never been studied experimentally. Specific high-quality data on *B. cereus* are still lacking, and so the differences and similarities remain to be studied in detail [1]. Of note, *B. cereus* is, evolutionary speaking, quite distinct from *B. subtilis* [4].

To gain new insights into the dynamic changes in the spore proteome during *B. cereus* spore germination and outgrowth, we conducted a high temporal resolution mass spectrometry analysis using label-free quantification. Our data show that the levels of germination associated proteins and morphological changes during the germination and outgrowth of *B. cereus* spores are consistent with the current germination and outgrowth model of *Bacillus*. The network analysis provided a global view of the protein level dynamic changes from eight clustered modules. Furthermore, proteins that may have functions in germination were predicted by the network analysis. From an applied point of view, our study revealed physiological processes during the awakening of dormant *B. cereus* spores and their return to vegetative life.

## 2. Results

### 2.1. Morphological and Proteomic Studies on Spores during Germination and Outgrowth

To verify whether the model of germination based largely on work with *B. subtilis* is applicable to *B. cereus*, a series of samples were collected in triplicate during *B. cereus* spore germination and outgrowth. The morphological changes in these samples were observed under the microscope and are shown in Figure 1b. It can be observed that the spores were in the bright phase before and after heat activation (PHA and AHA). After the addition of germinants and the transfer to the germination medium (0 min), the spores started germination immediately, transitioning from the bright phase to the dark phase. The whole population of spores finished germination within 10 min. After 10 min, upon entering the outgrowth, the spores swelled gradually and grew out. After completion of the germination and outgrowth, the spores reverted back to the actively dividing vegetative state and cell division was observed at 180 min. 

Next, we evaluated the spore germination and outgrowth of *B. cereus* at the protein level. For the proteomic study, in total, 2211 proteins were identified and quantified. The reproducibility of samples was assessed by calculating the Pearson correlation coefficients among three biological replicates at each time point. The pair-wise comparison of all of the time points displayed high precision correlations over 0.8 (Figure 2A). The replicates collected between AHA to 60 min were relatively well correlated. Moreover, the replicates collected during the vegetative growth phase (120–240 min) also possessed high correlations. To investigate the proteome differences between samples, log2 transformed LFQ intensities were used for the principal component analysis (PCA) and hierarchical cluster analysis. As shown in Figure 2B, the first principal component clearly differentiated most samples along the time axis. It was noteworthy that the PHA sample was distinct from the AHA sample in the second principal component, indicating significant changes in the measured proteome induced during heat activation. This difference was also observed in the hierarchical cluster analysis (Figure 2C).

The samples at each time point were compared with the T0 samples and the proteins that exhibited a significantly differential expression (Appendix A) were identified using the limma algorithm. According to the microscopy, spores eventually grew into vegetative cells and regained the ability to divide at T180. As shown in Figure 3, a comparison of the proteome at T240 and T0 revealed the differences between the spores and vegetative cells. We found 457 proteins were spore predominant and 284 proteins were vegetative cell predominant, and 743 proteins were shared between the spores and vegetative cells. Spore predominant proteins included spore coat proteins and germination associated proteins, as well as proteins that are important in spore resistance or serve as an amino acid source early in spore outgrowth. Vegetative cell predominant proteins are mainly involved in metabolism, including energy, amino acid, lipid, and nucleotide metabolism, as well as cell signal transduction and transport. The abundances of five proteins were significantly increased at T10, when germination was just finished, four of which were associated with transcription. The levels of 130 proteins were significantly changed at T60.

### 2.2. Dynamic Changes of Spore Specific Proteins

According to the current model, germination requires the involvement of GRs, channel proteins for CaDPA release, and proteins for the signal transduction and degradation of the cortex and part of the coat proteins [5]. A number of SASPs are also degraded as germination is completed, including those protecting the spore DNA from damage, and their degradation allows for DNA transcription. The degradation of the SASPs as well as that of other low molecular weight spore proteins also provides a pool of amino acids for initiation of the metabolism and protein synthesis; the latter can be especially important as spores lack all enzymes needed for amino acid biosynthesis [1,2,5]. The shedding and degradation of some spore coat proteins in outgrowth also facilitates spore expansion. The dynamic changes in spore specific proteins were quantified and are shown in Figure 4. The levels of all spore specific proteins dropped during the spore outgrowth (t = 60–180 min), which is consistent with studies in *B. subtilis* [3] and *B. anthracis* [6] spore germination and outgrowth. Between T0 to T30, the intensities of proteins present in the germination pathway (Figure 4A) and coat layer (Figure 4B) remained constant. The levels of proteins that are a part of the germination pathway (Figure 4A) declined gradually in quantity from 30 min onward, while the coat protein levels decreased from 60 min (Figure 4B). The level of the GerI GR’s A subunit (GerIA) fell to an extremely low level at T240, but the intensities of GerIC, GerD, SpoVAD, and SpoVAF showed relatively slow and gradual decreases. *B. cereus* spores had two paralogues of CwlJ. The level of CwlJ1 decreased to an extremely low level at T240, but the other one, CwlJ2, remained relatively high at 240 min. The intensity of CotH declined faster than the other coat proteins during outgrowth (60–120 min). The abundances of SASPs gradually decreased from the onset of germination, showing the biggest decline early in the outgrowth (10–30 min, Figure 4C). One noteworthy protein was the single gamma-type SASP (BC0506), which saw a continued rapid decline from 0 min. Studies have also shown that *B. subtilis* homologs of *B. cereus* SASP-1, SspF, SspH, SspI, SspP, and Tlp are also degraded early in spore germination. Notably, the protease, GPR, which initiates most SASP degradation, also disappeared in this period. In contrast, the abundances of exosporium proteins showed decreases only from T60, when the spores were well into the outgrowth. 

By ortholog mapping to *B. subtilis*, highly conserved global regulators were quantified in *B. cereus*, including SigA, AbrB, CodY, CcpA, and SpoVT and the multi-drug exporter Mta [7]. Compared with the level at 0 min, the Mta level significantly increased at 10 min just at the end of germination when the spore core was partly rehydrated (Figure 4F). The AbrB abundance increased dramatically between 10–30 min. The abundances of CcpA, SpoVT, and CodY were trending downward from 30 min overall [8], even though there were oscillations of CcpA between 60 to 180 min (Figure 4E). The abundance of SigA continued to decrease until 120 min, when it started to rise again (Figure 4G).

### 2.3. Co-Expression Network Construction

Proteomic data were further analyzed and visualized using weighted gene co-expression network analysis (WGCNA). WGCNA was developed in the field of systems biology to describe the correlations among the genes across microarray samples. The application of WGCNA analysis on proteomic datasets could be achieved by transforming the proteomic datasets into a format similar to the microarray and RNA-seq datasets after data normalization and imputation [9]. Proteins in a co-expressed module indicate a close regulatory relationship, which can be used to infer unknown protein functions by their association with a cluster of well-characterized proteins [10]. 

In this study, we constructed a signed weighted co-expression network based on proteomics data to unravel protein dynamic patterns during spore germination and outgrowth and to identify uncharacterized proteins behaving similarly to well-known germination proteins, which may be inferred to also have important functions during germination. Germination proteomics data were imputed to avoid issues caused by missing values before the network construction. As it was observed that heat activation had a large impact on the dormant spore proteome, the PHA and AHA samples were eliminated from the network construction to remove the influence caused by heat activation on the module construction. To improve the co-expression network analysis, a high-quality dataset of 1175 proteins identified in at least two replicates at each time point (except PHA and AHA) was used for further analysis. In total, eight modules (clusters) were identified (Appendix A): “black” (41), “blue” (133), “brown” (131), “green” (87), “grey” (454), “red” (53), “turquoise” (153), and “yellow” (123). Proteins that did not show any similarity to other modules were clustered into the “grey” module. The protein correlation in the network was shown in a heatmap (Appendix A). The correlation of each module was represented using an Eigenprotein, which is defined as the first principal component in one module and can be considered as a representative of the protein profiles in a module [11], and is shown in Appendix A. To understand the relationships between the modules and the spore germination and outgrowth processes, the modules were correlated with time points, as is depicted in Appendix A. All modules had weak coefficients of correlation with germination and early outgrowth processes (0–60 min). The “blue” module had a strong negative correlation with T240. The “brown” module was significantly negatively correlated with T180.

The changing trends for the modules are shown in Figure 5. Proteins in the “brown” module exhibited a quick decrease in intensity from 60 to 120 min, and then remained constant. The protein levels of the “blue” module decreased gradually from 60 min. The abundances of proteins in the “green” module slightly increased from 10 min when the spores first finished germination and peaked at 120 min. The amounts of proteins in the “red” and “black” modules decreased from 60 min and showed an increase at 240 min. The protein levels in the “turquoise” and “yellow” modules increased dramatically at 120 min. In the “turquoise” module, the protein level maintained an upward trend, but the “yellow” module proteins were relatively stable after 120 min. To check whether modules were clustered with biological relevance, function enrichment analysis was performed (Appendix A). The enrichment scores (FDR) suggested that modules were enriched with biological meaning.

We examined each module and found that proteins that are known to play important roles in germination and outgrowth were clustered into “brown” and “blue” modules (Appendix A). As stated above, the “blue” module was significantly negatively correlated with T240 and the “brown” module was significantly negatively correlated with T180. To further screen for potential key proteins in the “brown” and “blue” modules, we calculated the correlations between the proteins at T180 and T240 (gene significance (GS)), and the correlations between the proteins in the “brown” and “blue” modules (module membership (MM)). Proteins with a high connectivity within a module (weight ≥ 0.3), correlations with T180 in the “brown” module or with T240 in the “blue” module (GS ≤ −0.5), and correlations with modules (MM ≥ 0.8) were considered to be potential key proteins. In the “brown” module, 59 proteins were sorted out to be potential key proteins and 65 were sorted in the “blue” module. The top regulons for these key proteins were SigE, SigA, SigK, and SigG, as expected for the spore coat and germination proteins.

## 3. Discussion

Under natural conditions, spores germinate, albeit heterogeneously, upon encountering nutrient molecules in the environment, such as certain amino acids, sugars, and purine nucleosides. Heat activation is a sub-lethal short thermal treatment that enhances and synchronizes spore germination. As a tool to facilitate the subsequent biochemical analysis of germinating and outgrowing spores, heat activation procedures are widely used in spore germination and outgrowth studies in laboratory settings [12,13]. A previous study in *B. cereus* reported that L-alanine did not induce significant germination without heat activation [5]. The temperature and heating time used in this study for activating *B. cereus* spores was adopted from a previous study [14] in order to obtain the activation effects on the spores rather than causing detectable damage. However, we observed significant effects of heat activation on the measured proteome of *B. cereus* spores. The reasons for the differences in the proteome with or without prior heat activation are still unknown. Such differences may have to do with changes in the cellular biophysical state [15], influencing protein structure and extractability and resulting in a large change in detectable proteins. A reasonable assumption is that sub-lethal heat may change the structure of spores, i.e., their inner membrane or coat, which affects protein extractability and thus the spore proteome measured. The biophysical effects of spore heat activation, pertaining to the spore core state, were observed previously [15].

One aim of this study was directed towards a global view of the germination and outgrowth processes at the level of the proteome in *B. cereus* spores. The observations were consistent with the current germination model in *Bacillus* species. Although general morphological changes of *B. cereus* spores in the germination and outgrowth followed similar steps as the *B. subtilis* spores [3], *B. cereus* spores germinated much faster. In germination (T0-T10), the water content in the spore core increased gradually. The abundances of most proteins remained constant, which was different from the proteomic study during *B. subtilis* spore germination, where obvious protein decreases of a number of proteins were detected upon germination initiation [3]. Nevertheless, the effects of protein synthesis in early germination were observed. The abundances of the global regulators of AbrB and Mta increased during the early phases of germination (T3-T10) and significant abundance changes were detected at 10 min. One SASP decreased during the whole germination and outgrowth phases. AbrB, as a key transition state regulator steering vegetative growth, has been studied in depth in *B. subtilis* [16], although the specific function of AbrB in *B. cereus* spore germination is unknown. Mta is a multidrug-efflux activator [17] and it may contribute to the resistance of the germinating spore to environmental killing factors. During early outgrowth (T10-T60), proteins in the germination signal transition pathway, such as cortex lytic enzyme CwlJ and channel protein SpoVA, started to degrade. Protein levels in the “green” module were slightly increased and are involved in translation and cell wall synthesis. During later outgrowth (T60-T180), the abundances of proteins in the “black” and “red” modules significantly increased, suggesting increased activity in the protein synthesis and energy metabolism. The coat proteins were rapidly degraded during this period to facilitate spore swelling in the outgrowth. The noticeable degradation of CotH suggested that CotH may be a fundamental protein for the coat layer, and its degradation may be a key step in coat shedding. It has been reported that the coat protein CotH is located in the inner coat as a morphogenetic factor mediating the assembly of at least nine coat proteins [18,19,20]. In *B. subtilis,* CotH is reported to play a role in germination [21].

For the main sigma factor, SigA, which is mostly responsible for the transcription of housekeeping genes in the exponential growth phase [22], its abundance decreased after the spores were fully hydrated (T10-T120). Even though regulators are conserved, the roles they play in two organisms can be different. For example, SpoVT dramatically affects the completion of sporulation and the deletion of *spoVT* causes premature germination of pre-spores in *B. cereus*, but it is not essential in sporulation of *B. subtilis*, and rather is associated with spore germination heterogeneity [23]. 

Label-free quantification is widely applied in proteomics, as it is widely applicable on varied sample types and allows for the comparison of large numbers of samples. In addition, label-free quantification is economic and simple, without extra steps of introducing stable isotopes during sample preparation. However, the label-free method cannot monitor new protein synthesis until abundance changes are detected. Further examination of the onset of protein synthesis can be monitored by using a pulsed labeling method, such as dynamic SILAC (stable isotopic labeling of amino acids in cell culture). The SILAC incorporation experiments in *B. subtilis* reviving spores showed that this method is sensitive to detect the onset of the synthesis of low abundance proteins during spore germination [3].

Protein network analysis provides valuable data to study proteins with similar profiles based on a “guilt-by-association” approach [24]. The basis of this network analysis allows for studying protein behavior patterns, as well as the identification of novel protein functional interrelationships and correlations to germination and outgrowth stages. However, to reduce the influences of data quality, only proteins that were identified in every time point were used to generate the co-expression network. The proteins that were completely degraded or newly synthesized would not be analyzed using WGCNA in this way. The potential key proteins analysis based on network connections (weight ≥ 0.3) and GS and MM led to 59 proteins in the “brown” module and 65 proteins in the “blue” module with high possibilities of having putative roles in germination and outgrowth. It is worth noting that 17 proteins were uncharacterized proteins. The length of all uncharacterized proteins was longer than 100 amino acids, indicating that none of them were likely to be SASPs. It will be very interesting to study the functions of these 17 proteins in spore germination and outgrowth in *B. cereus* spores in the future.

In summary, we successfully applied WGCNA to spore germination and outgrowth proteomic data. The analysis showed that there were a number of modules that were clearly correlated with protein functionality during *B. cereus* spore germination and outgrowth. Future studies using genetic approaches, where mutants in the identified genes with novel putative roles in germination and outgrowth will be interrogated, are needed. Such approaches should provide more definitive functional data, allowing us to draw conclusions about the identification of the functional protein modules that emerge during the spore germination of the food borne toxigenic organism *B. cereus*. Such protein modules may be cause or consequence, and hence follow or steer germination and outgrowth.

## 4. Materials and Methods

### 4.1. Morphological and Proteomic Studies on Spores during Germination and Outgrowth

*B. cereus* strain ATCC 14,579 was used in this study. The details of the spore culture and harvest were described previously [13,25]. In brief, a single colony was inoculated in tryptic soy broth (TSB) and grown aerobically at 30 °C overnight. Then, the vegetative cells were collected by spinning down and they were cultured in a chemically defined growth and sporulation (CDGS) medium for 96 h to accomplish sporulation. The spores were harvested by centrifugation and washing with cold Milli-Q water at least four times, stored at 4 °C, and were used for the experiments within three days

Fresh spores were heat activated at 70 °C for 15 min before germination. Around OD600 = 2 (final concentration) spores were germinated in 100 mL YE medium at 30 °C with the germinants. The final concentrations of germinants were 10 mM L-alanine and 0.1 mM inosine. The germination of *B. cereus* spores was a fast process that took around 5–10 min. Samples were taken before and after heat activation and subsequently during germination and outgrowth as the various phases of spore revival (Figure 1) emerged. Overall, it took the heat-activated spores ~ 10 min to complete germination, and over 2 h to outgrow into vegetative cells, so the interval times were longer between the later sampling time points. In summary, one ml of sample was taken at PHA (Pre-heat activation), AHA (After heat activation), and then at 0 min, 3 min, 5 min, 10 min, 30 min, 60 min, 120 min, 180 min, and 240 min. The samples were quenched with a final concentration of 20% cold methanol and kept in liquid nitrogen for short storage. Three biological replicates at each sampling time were obtained and stored at −80 °C. The YE medium [25] was composed of 0.00005% FeSO4·7H20, 0.0005% CuSO4·5H20, 0.0005% ZnSO4·7H20, 0.005% MnSO4·H20, 0.02% MgSO4, 0.008% CaCl2·2H2O, 0.05% K2HPO4, 0.2% (NH4)2SO4, 0.2% yeast extract, and 0.1% glucose. Stocks of all minerals were prepared individually and mixed before use.

### 4.2. Sample Preparation and LC-MS/MS Analysis

All of the samples were processed using the “one-pot” method, as described previously [26]. The resulting peptides were resuspended in 0.1% formic acid and around 200 ng was injected into the Bruker tims-ToF. Mass spectrometry analysis of all samples was performed with tims-TOF pro (Bruker) coupled with an Ultimate 3000 RSLCnano UHPLC system (Thermo Scientific, Germeringen, Germany). The peptides were injected onto a 75 µm × 250 mm analytical column (C18, 1.6 µm particle size, Aurora, Ionopticks, Australia) and were separated by a multi-step gradient of mixtures of solvent A (0.1% formic acid in water) and B (0.1% formic acid in acetonitrile): 55% solvent B for 55 min, 21% solvent B for 21 min, 31% solvent B for 12 min, 42.5% solvent B for 3408 min, and 99% solvent B for 7 min. The parameters of the mass spectrometer were set to a spray voltage of 1.5 kV, a source gas flow of 3 L/min of pure nitrogen, a dry temperature setting of 180 °C, PASEF mode of acquisition, a scan range between 100–1700 m/z, and a tims range of 0.6–1.6 V.s/cm^2^. Fragmentation spectra were generated with 2 Th at 700 m/z and 3 Th at 800 m/z quad isolation width, and collision energy from 20 to 59 eV. A total of 10 PASEF MS/MS scans were scheduled with a total cycle time of 1.16 s.

### 4.3. Data Analysis

The acquired LC-MS/MS data were processed with Maxquant (ver.1.6.14.0) [27] software and searched against the *B. cereus* ATCC 14,579 database downloaded from Uniprot (downloaded in September 2019). Trypsin/P was set as the digestion enzyme, and a maximum of two missing cleavages were allowed. The carbamidomethylation (C) was set as a fixed modification and oxidation (M) as a variable modification. The option “Match between runs” was selected with a matching time window of 0.2 min and a matching ion mobility window of 0.05 indices. Both iBAQ and LFQ were used for the label free quantification. Proteins identified in at least two replicates were considered to be valid. Then, the data were normalized and imputed in Perseus (version 1.6.15.0) [28] using LFQ intensity, as well as the analysis of PCA and the cluster heatmap. The function enrichment was performed on DAVID (database for annotation, visualization, and integrated discovery). For the orthology analysis, OrthologueDB [29] and OMA browser [30] were used. Significantly differentially expressed proteins that were quantified in at least 24 of 36 samples were analyzed using R package limma [31]. Log2 transformed LFQ abundances were fitted to a linear model and the standard errors were smoothed by empirical Bayes. Proteins with a|log2(fold change)|≥ 1 and adjusted *p*-value ≤ 0.05 were considered to be significantly expressed. The R package WGCNA was applied for network analysis [9,11], and the network was constructed based on a signed correlation. The parameter soft-threshold power *β* was carefully chosen to lead to an approximate (R2 ≥ 0.9) scale-free topology network. The WGCNA modules and eigenproteins (the first principal component of the modules) were identified and are representative of the whole modules. Furthermore, modules were correlated to the time points (traits) and those with *p*-value < 0.05 were considered significant.

## Figures and Tables

**Figure 1 ijms-23-13614-f001:**
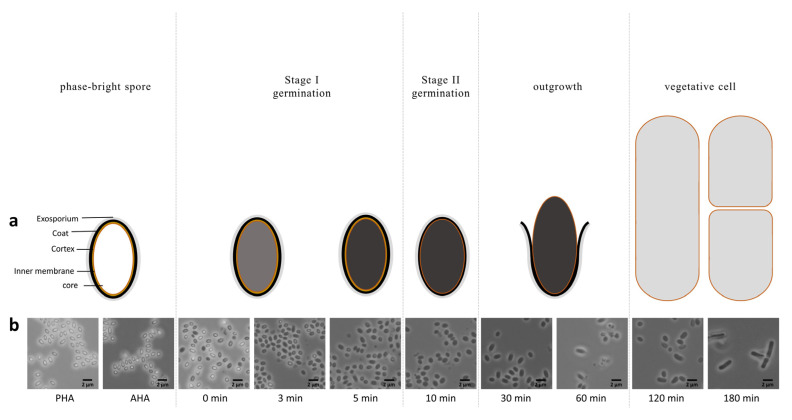
Scheme of spore germination and outgrowth (**a**). Morphological changes taking place during spore germination and outgrowth, as seen by microscopy (**b**). Spore samples were harvested at different time points throughout germination and outgrowth, and were imaged by phase contrast microscopy. PHA, pre-heat activation; AHA, after heat activation.

**Figure 2 ijms-23-13614-f002:**
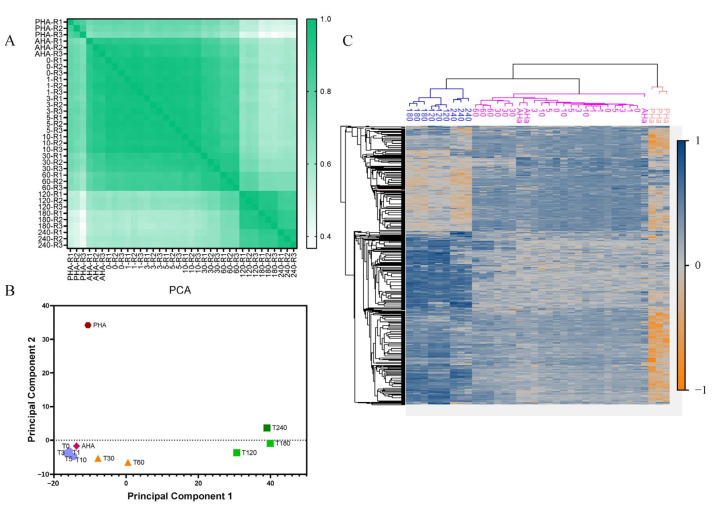
Comparison of the proteome at different time points during *B. cereus* spore germination and outgrowth. (**A**) Pairwise Pearson correlations between all replicates. (**B**) PCA score plot of germination at different time points. (**C**) Cluster heatmap of the proteome-wide analysis. The color represents the z-score transformed protein amounts of the LFQ intensity. Blue corresponds to an increased expression level, grey corresponds to an average level, orange corresponds to a decreased level, and columns and rows represent replicates at different time points and proteins, respectively. The time points are as follows: pre-heat activation (PHA), after heat activation (AHA), 0 min (T0, addition of germinants), 1 min (T1), 3 min (T3), 5 min (T5), 10 min (T10), 30 min (T30), 60 min (T60), 120 min (T120), 180 min (T180), and 240 min (T240).

**Figure 3 ijms-23-13614-f003:**
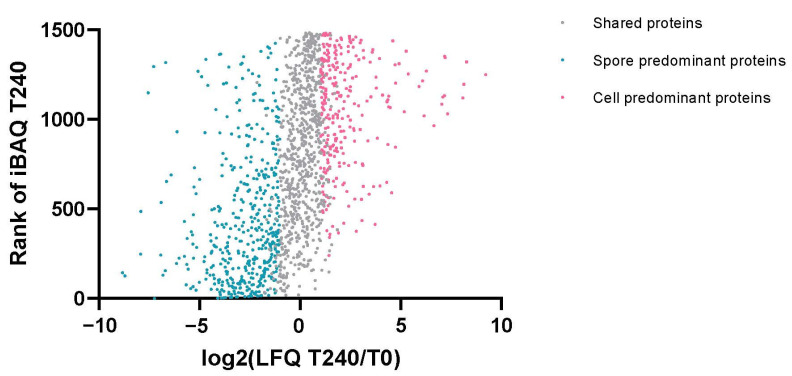
Scatter plot of the proteome comparison of the spore and vegetative cells. The *y*-axis was the protein rank according to the iBAQ intensity at T240 (vegetative cells). Proteins with higher iBAQ intensities at T240 ranked lower. The *x*-axis is the log2 transformed LFQ intensity ratio of T240 and T0. The predominant proteins were determined using the limma algorithm.

**Figure 4 ijms-23-13614-f004:**
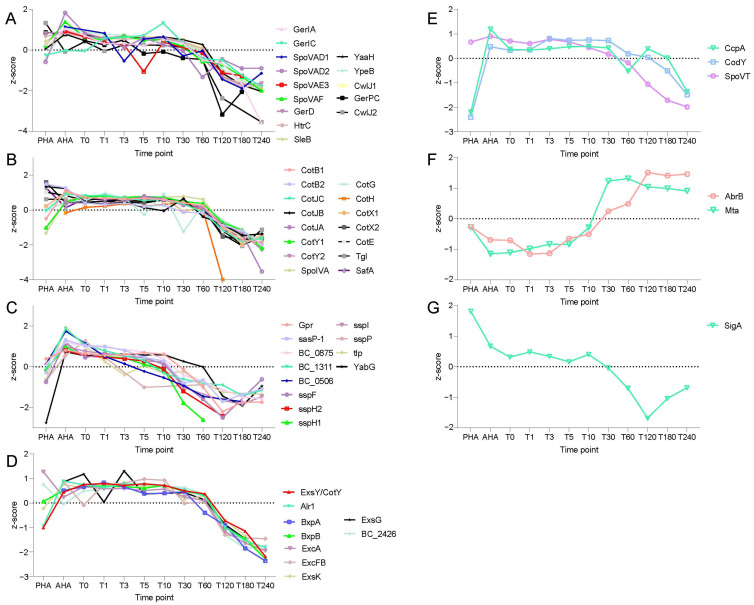
Dynamic changes in spore specific proteins during germination and outgrowth. Z-score profiles of the identified spore proteins across all of the times points are shown. (**A**–**D**) Identified germination pathway proteins, coat proteins, SASP, and exosporium proteins, respectively. (**E**–**G**) The abundance variations of conserved regulators at different time points. The Y axes display the z-scores transformed protein LFQ signal intensity and the *x* axes show the time points.

**Figure 5 ijms-23-13614-f005:**
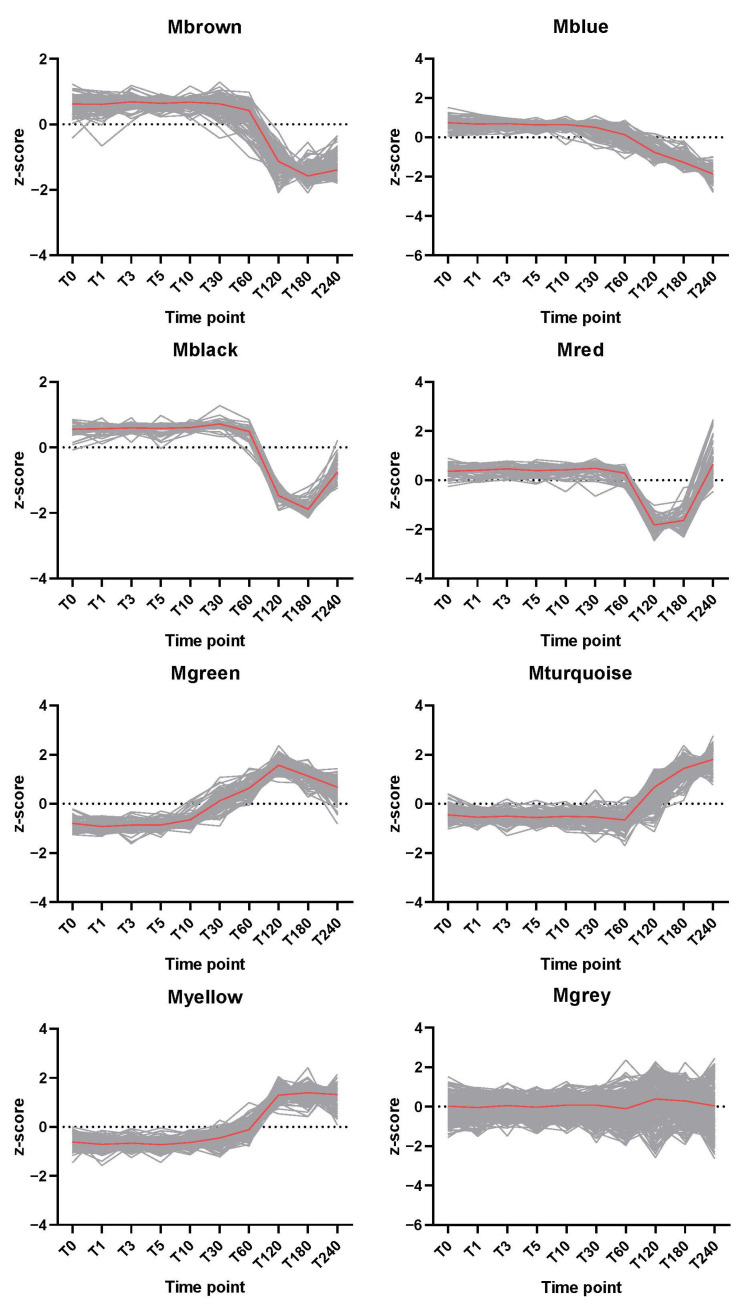
The trends in the z-score transformed protein levels of the modules across time points. The grey lines indicate protein level changes. The red lines represent the median z-scores for each module.

## Data Availability

Mass spectrometry data have been deposited and can be found at ProteomeXchange (PXD036146), and the Massive Repository for Mass Spectrometry data (MSV000090154).

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
