# Peer review of "Time-Resolved Proteomics of Germinating Spores of Bacillus cereus"

_ijms, 2022, doi:10.3390/ijms232113614_

Round 1

Reviewer 1 Report

The general writing is clear and contains sufficient background to highlight the relevance of the study, which describes the protein profiles at different stages upon germination of the spores. This is also relevant as there are few studies comprising proteomic-level analysis in the developmental stages of spores for the species B. cereus.

In general, the experiments are well designed, and the results clearly described in terms of answering the question of getting ‘ new insights into dynamic changes in the spore proteome’. 

I have no further concerns or inaccuracies that might affect the acceptance of the article.

Reviewer 2 Report

The article "Time-resolved proteomics of germinating spores of Bacillus cereus" reports the dynamics of protein expressions in Bacillus cereus during spore and vegetative state. The idea is very interesting and the importance of this findings cannot be ignored.

Major comments:

- The importance of cells used dor this studies and if possible cell cycle passages can be explained in the materials and methods for clear understanding the cell cycle (age) of the cultures used for performing all the experiments.

- The expression of differentially expressed proteins at different stages as well as states of Bacillus cereus growth can be utilised to further explain the protein-protein interactions and their association with the process explained to get a better understanding about the effects involved in each cases.

- Since, comparison between proteiomics of both spore and vegetative cells have been done to determine the commonly and differently expressed proteins along with co-expression network analysis. Can authors derive any conclusions or propose any hypothesis on spore survival and/or revival.

- Considering the previously published article by the same authors:

Gao, X.; Swarge, B.N.; Roseboom, W.; Wang, Y.; Dekker, H.L.; Setlow, P.; Brul, S.; Kramer, G. Changes in the Spore Proteome of Bacillus cereus in Response to Introduction of Plasmids. Microorganisms 2022, 10, 1695. https://doi.org/10.3390/ microorganisms10091695

it is very important to explain the association of such changes with plasmid or any other factors (like bacteriophages, etc.) to properly attribute such morphological as well as proteiomic changes observed during this studies.

Minor comments:

- Figures must be uniformly modified (fig 4).

- Other articles can be cited in support of the conclusions 

Reviewer 3 Report

Reviewer comments on IJMS manuscript "Time-resolved proteomics of germinating spores of Bacillus cereus" by Xiaowei Gao et al.

In their manuscript " Time-resolved proteomics of germinating spores of Bacillus cereus” the authors describe the results of a time-resolved proteomics study of B. cereus spore germination following heat activation. At time points  from 0 to 180 minutes, spores are lysed and their proteome evaluated by bottom-up mass spectrometry, database searching and label-free quantitation. Following consistency checks, 1175 proteins constituting a ‘high quality dataset’ are evaluated for significant up- and downregulation, as well as for time-course clustering and weighted gene co-expression network analysis (WGCNA). The authors describe a total of eight clusters showing similar temporal profiles, mostly with inflection points between T30 and T60 during outgrowth phase. No additional experimentation beyond the mass spectrometry analysis is described in the manuscript.

Quality. The manuscript is overall of good technical quality and detail. Text and figures are clear and well-structured, the data and the authors’ arguments laid out clearly. The quality of experimentation is hampered by a lack of experimental replication (see below). The discussion of the identified protein clusters is largely descriptive and lacks detail.

Relevance and novelty. The manuscript uses a methodology previously used by the same authors to study spore composition and germination in Bacillus subtilis and Bacillus cereus (doi.org/10.3390/ijms222212475, doi.org/10.3390/microorganisms10091695). While Bacillus cereus has significant importance as a widespread food pathogen and therefore an understanding of its spore germination process would be of importance, little novel biological detail is revealed or discussed by the authors in the manuscript.

There are a number of issues which keep me from recommending the manuscript from publication:

1)      Lack of corroborating experimentation (major). The manuscripts solely describes the results of a label-free mass spectrometry experiment, with the addition of a few microscopic images. There is no experimentation by independent experimental approaches, even as basic as protein quantity determination or SDS-PAGE profiling along the time course, or by e.g. antibody- or fluorescence-based methods (which are established in the authors’ labs, see doi.org/10.3390/microorganisms10091695). There are also no functional experiments described here, nor experiments including genetic experimentation (which the authors clearly propose themselves at the end of section 3). Thus, the mass spectrometric results have to be taken at face value.

2)      Lack of replication (major). The mass spectrometric experiments were performed at n=3 total, without technical replication on the MS level or higher replication on the biological level. While the graphical evaluation and time course plotting seems to provide compelling results, this is clearly insufficient to support significance testing as performed and described by the authors. This is even more striking since no independent experimental approaches were used to validate at least subsets of proteins, and no means are provided to evaluate the level of reproducibility of the data (such as t-Testing results, Volcano plots etc.). It is thus impossible for reviewers and readers alike to judge the quality of the data.

3)      Lack of detail in the discussion (major). The key part of the results is clearly the construction of co-expression networks. Here, the authors identify 8 co-expression clusters which are denominated by colours “black”, “blue” etc. The constituent proteins of each cluster, however, are not analyzed or discussed in much detail beyond a few ‘househould’ protein examples, which are however not even highlighted in the co-expression cluster graphs (Fig. 5). As such, the discussion of the analysis is superficial and does not reveal novel biological insight.

4)      No reviewer details for raw data access provided (minor). The manuscript does contain ProteomeExchange and Massive Access data, however no reviewer access is granted.

5)      Slight exaggeration of the results (minor). In the abstract, the authors report ‘monitoring 2211 protein changeovers’. The actual ‘high quality’ data set that is used for co-expression clustering, however, is reduced to 1175 proteins for the sake of consistency. The authors should be more clearly about this throughout abstract and summary,

Overall, the manuscript is missing significant detail to be recommended for publication as a full research article in IJMS, I am afraid. Elevating the manuscript to this level would require significant additional experimentation as outlined above, including replication of the core mass spectrometric data set. In addition, the content of the manuscript is not a good thematical fit for the IJMS section Molecular Genetics and Genomics. As such, I recommend rejection of the manuscript, however would encourage the authors to resubmit a more complete set of experimentation on the same topic in the future. Alternatively, they may consider publishing the existing manuscript as a data set brief in a dedicated proteomics journal.

Round 2

Reviewer 3 Report

While I would like to thank the authors for revising their manuscript, I can unfortunately not see that some of the major shortcomings highlighted in the first round of review were addressed. 

In particular, there are still no corroborating experiments using independent methods, there is still no functional delineation of the mass spectrometric data, and the (still compelling) time-course clustering of the data still remains largely unexploited.

In addition, I would still question the use of significance testing based on n=3. I indeed do have significant experience with peptide and protein mass spectrometry (25 years) and with both performing and reviewing quantitative proteomics data (10 years). Based on this experience more than anything, I would stress the requirement for independent experimental approaches and sound statistics even more. The authors' suggestions that their current approach was accepted by another journal previously, or that mass spectrometry-based proteome profiling is a 'gold standard method' anyway, do not make the shortcomings any better. The raw LFQ values are indeed available as part of the supplementary data, however the authors have not put these into a format  that makes them accessible for readers and reviewers. It is not sufficient to just present the 'raw' data and expect readers to do their own evaluation and draw their own conclusions.

Since the manuscript is severely limited in both these key requirements, I would strongly recommend that the authors publish the data e.g. as a data set brief in a dedicated proteomics journal, e.g. J Proteome Res. This would still enable the research community to benefit from the data. Unfortunately, IJMS does not offer data set briefs as a publication format. As the manuscript does not meet the standards for a full proteome study - which the authors implicity acknowledge by pointing to future experimentation future publications - I unfortunately still cannot recommend the manuscript for publication in IJMS.